# The Hep-CORE policy score: A European hepatitis C national policy implementation ranking based on patient organization data

Adam Palayew[1‡], Samya R. Stumo[2†‡], Graham S. Cooke[3], Sharon J. Hutchinson[4], Marie Jauffret-Roustide[5], Mojca Maticic[6,7], Magdalena Harris[8], Ammal M. Metwally[9,10], Homie Razavi[11], Jeffrey V. Lazarus[2]*, on behalf of the Hep-CORE Study Group

1 McGill University Department of Epidemiology, Biostatistics, and Occupational Health, Montreal, Quebec, Canada, 2 Barcelona Institute for Global Health (ISGlobal), Hospital Clínic, University of Barcelona, Barcelona, Spain, 3 Division of Infectious Diseases, Imperial College, London, United Kingdom, 4 School of Health and Life Sciences, Glasgow Caledonian University, Glasgow, United Kingdom, 5 Cermes3 (Inserm U988/CNRS UMR 8211/Ecole des Hautes Etudes en Sciences Sociales/Paris Descartes University), Paris, France, 6 Clinic for Infectious Diseases and Febrile Illnesses, University Medical Centre Ljubljana, Ljubljana, Slovenia, 7 Faculty of Medicine, University of Ljubljana, Ljubljana, Slovenia, 8 Department of Public Health, Environments and Society, London School of Hygiene & Tropical Medicine, London, United Kingdom, 9 Community Medicine Research Department, Medicine Research Division, National Research Centre, Giza, Egypt, 10 Association of Liver Patient Care, Dakhlyia, Egypt, 11 Center for Disease Analysis Foundation, Lafayette, CO, United States of America

† Deceased.
‡ These authors share first authorship on this work.
* Jeffrey.lazarus@isglobal.org

**Data Availability Statement:** The data and code are available at the following link https://osf.io/mh79q/.

## Abstract

### Background

New hepatitis C virus (HCV) treatments spurred the World Health Organization (WHO) in 2016 to adopt a strategy to eliminate HCV as a public health threat by 2030. To achieve this, key policies must be implemented. In the absence of monitoring mechanisms, this study aims to assess the extent of policy implementation from the perspective of liver patient groups.

### Methods

Thirty liver patient organisations, each representing a country, were surveyed in October 2018 to assess implementation of HCV policies in practice. Respondents received two sets of questions based on: 1) WHO recommendations; and 2) validated data sources verifying an existing policy in their country. Academic experts selected key variables from each set for inclusion into policy scores. The similarity scores were calculated for each set with a multiple joint correspondence analysis. Proxy reference countries were included as the baseline to contextualize results. We extracted scores for each country and standardized them from 0 to 10 (best).

### Results

Twenty-five countries responded. For the score based on WHO recommendations, Bulgaria had the lowest score whereas five countries (Cyprus, Netherlands, Portugal, Slovenia, and

**Funding:** The European Liver Patients' Association received funding from AbbVie, Gilead Sciences and MSD to carry out this study. GC is supported in part by the BRC of Imperial College NHS Trust and an NIHR Professorship and that JVL is supported by a Spanish Ministry of Science, Innovation and Universities Miguel Servet grant (Instituto de Salud Carlos III/ESF, European Union (CP18/00074)). JVL further acknowledges support from the Spanish Ministry of Science, Innovation and Universities through the "Severo Ochoa Centre of Excellence 2019-2023" Program (CEX2018-000806-S), and support from the Government of Catalonia, Spain, through the CERCA Program. ICMJE forms have been provided by all authors. The funders had no role in study design, data collection and analysis, decision to publish, or preparation of the manuscript.

**Competing interests:** The authors have read the journal's policies and declare the following competing interests: This study was directly funded by ELPA, who in turn were funded by AbbVie, Gilead Sciences, and Merck & Co. This does not alter our adherence to PLOS ONE policies on sharing data and materials.

Sweden) had the highest scores. For the verified policy score, a two-dimensional solution was identified; first dimension scores pertained to whether verified policies were in place and second dimension scores pertained to the proportion of verified policies in-place that were implemented. Spain, UK, and Sweden had high scores for both dimensions.

## Conclusions

Patient groups reported that the European region is not on track to meet WHO 2030 HCV goals. More action should be taken to implement and monitor HCV policies.

## Introduction

Viral hepatitis is one of the leading causes of death and disability worldwide and, since 2013, has surpassed HIV/AIDS and malaria in terms of annual deaths. An estimated 71 million people worldwide live with chronic hepatitis C virus (HCV) infection, including 5.6 million people in the World Health Organization (WHO) European Region [1]. The largest burden of HCV-related mortality is caused by liver cirrhosis and liver cancer [2]. In Europe, the populations most impacted are people who are incarcerated, people living with HIV (PLHIV) men who have sex with men (MSM), and people who inject drugs (PWID) [3]. According to the European Centre for Disease Prevention and Control (ECDC), in the European Union (EU)/ European Economic Area (EEA), 45.5% of new cases of hepatitis C with transmission information were attributed to injection drug use [4].

In 2016, WHO set the ambitious goal of eliminating hepatitis B and C as a public health threat by 2030 in its first viral hepatitis global health sector strategy (GHSS) [5]. Elimination of viral hepatitis is defined as a 65% reduction in mortality and a 90% (80% for HCV) reduction in the incidence of viral hepatitis [5]. This goal was soon followed by further specified targets in the WHO European Region's *Action plan for the health sector response to viral hepatitis* [6]. Despite these commitments, many countries have been slow to implement policies to achieve elimination targets.

Prior to the GHSS, two global viral hepatitis policy monitoring studies were conducted and highlighted the importance of addressing the viral hepatitis epidemic. These studies were inspired by the absence of a European or global framework to monitor viral hepatitis policy and the increasing availability of all-oral, highly effective direct acting antivirals (DAAs) [7, 8]. To advance necessary civil society engagement and to fill the knowledge gap surrounding policy, the European Liver Patients' Association (ELPA) commissioned the Hep-CORE study in 2015 as a longitudinal project to monitor viral hepatitis policies in countries where they had member patient organizations. The Hep-CORE study works with patient groups to monitor policy implementation, traditionally the role of governments and global bodies. A notable exception to this is the community response to HIV, with civil society organizations, including patient groups, playing a pivotal role in the Global AIDS Response Progress Reporting. This has strengthened formal and informal monitoring functions, which has provided an inclusive perspective to compare data against reports issued by governments to global governance bodies, including WHO and UNAIDS [9]. Notably, civil society, and HIV community activism and involvement specifically, have improved policy monitoring, highlighted the needs of marginalized groups, and fostered effective government and civil society collaboration [10].

In its first two years, Hep-CORE focused on patient group reporting on the existence of viral hepatitis policies in their country. Results at baseline (2016) showed that there were few

policies supporting viral hepatitis policy recommendations, and, in 2017, only 52% of European countries had a national viral hepatitis plan or strategy in place [11–13]. In 2018, the Lancet Gastroenterology Hepatology Commission called for measuring national responses to viral hepatitis to ascertain a country's progress towards its elimination [14]. Inspired by this call, Hep-CORE 2018 set out to assess whether known national HCV policies in Europe, the United Kingdom (UK), and the Mediterranean Basin are functioning in practice and to score the extent of implementation according to patient advocacy groups.

## Methods

### Data collection

Participants were recruited through a purposive sampling process. We emailed an invitation to participate in the survey to one liver patient group in each of the countries where ELPA had members at the time of 2018 study recruitment. In countries with more than one patient group, we selected the group that was most involved in viral hepatitis advocacy in collaboration with the President of ELPA. A random sample of participants was not possible as the number of potential participants per country was often one and at most three. We invited the participants to complete the survey online using a customised survey link auto-generated by the Research Electronic Data Capture (REDCap) system, hosted online by Rigshospitalet, University of Copenhagen, Denmark, and distributed to each individual via email. Data were collected in September–October 2018. After data collection, we reviewed data and queried study participants via email about incomplete, inconsistent, or unclear information. Demographics and specific detail of who responded were collected, but are not disclosed to protect the anonymity of the respondents.

The survey questions were closed-ended and used a 5-point Likert scale [15]. A free-text box was available for participants to further comment on their responses. Additionally, to assess the base knowledge of the study participants, a 4-point scale appeared after each question asking the participant about their level of knowledge pertaining to the question above. We developed and revised the survey in accordance with multiple rounds of input from the members of the multidisciplinary study group, which included patient group representation. The study instrument was piloted in September 2018 with four patient groups from Egypt, Finland, Slovenia, and the UK and revised accordingly. We created and managed the final survey using REDCap.

We surveyed 30 patient groups each representing one country in the EU, Mediterranean basin, and the UK (in the EU at the time of the study). The study instrument comprised 23 questions derived from previous Hep-CORE surveys, peer-reviewed publications, verified data sources, expert group consultation, and the Global Health Sector Strategy (GHSS) [5, 16–20]. A full-length copy of the questionnaire is available in appendix 1. One community representative from each organization was responsible for completing the survey in collaboration with the organization´s board members.

The survey comprised two sets of questions: those based on WHO recommendations from the GHSS and those based on verified data sources like the European Monitoring Centre for Drugs and Drug Addiction (EMCDDA). The first set of questions (WHO recommendations) were provided to all countries and were based off the GHSS to assess implementation of WHO recommendations. The second set of questions (verified policy) were only sent to countries that had validated data sources confirm the presence of policies to assess implementation of these existing policies (Table 1). The data sources for the verified policies were selected as they either reported novel, raw data that informed the question that it was verifying or because they were reports from organizations that were responsible for collecting the monitoring data that underlie specific questions.

**Table 1. Main HepCORE 2018 survey questions.**

| Main question (number of sub questions) | Number of countries where question is relevant (1–25) | Data source |
|---|---|---|
| Elimination efforts HCV (NA) | 25 | Polaris/GHSS |
| Policy barriers equity (NA) | 25 | GHSS |
| Policy stigma and discrimination (NA) | 25 | GHSS |
| Policy HCV strategy/action plan (NA) | 23 | GHSS/Hep-CORE 2017 [11] |
| Policy HCV strategy/action plan comprehensive (6) | 11 | GHSS |
| Non-hospital testing (NA) | 12 | HIV in Europe unpublished data |
| Non-hospital treatment (NA) | 5 | Marshall 2018a [18] |
| Non-specialist treatment (NA) | 1 | Marshall 2018a [18] |
| Monitoring implementation (NA) | 25 | GHSS |
| Integration of HCV services with existing care (9) | 25 | GHSS |
| Targeting of special populations for elimination (14) | 25 | GHSS |
| Fibrosis HCV treatment restrictions (NA) | 12 | Marshall 2018b [17] modified to add Denmark |
| Drug and alcohol treatment restriction (NA) | 18 | Marshall 2018a [18] |
| Needle syringe programme general population (NA) | 21 | EMCDDA |
| Opioid substitution therapy general population (NA) | 21 | EMCDDA |
| Needle syringe programme prison (NA) | 3 | Bielen et al. 2018 [16] |
| Opioid substitution therapy prison (NA) | 16 | EMCDDA |
| Testing and screening for HCV in prisons (NA) | 16 | Bielen et al. 2018 [16] |
| HCV treatment in prisons (NA) | 16 | Bielen et al. 2018 [16] |

Questions derived from the GHSS were received by all participants, whereas the other data sources are the verified policy questions and were received only by selected countries that had the verified policy in place. Questions derived from EMCDDA data indicate that the information is derived from the country profiles of the European Monitoring Centre for Drugs and Drug Addiction.

## Data analysis

We descriptively and geospatially analysed final data using R 3.6.2 and Microsoft Excel 2017 version 15.31 for storage. Frequencies and count summaries were carried out for every question for all of the possible respondents. We further estimated two sets of similarity scores that we describe below.

**Policy similarity score estimation.** In addition to descriptive statistics, we used the multiple correspondence analysis (MCA), a dimension reduction method to evaluate the similarity between patient group responses from different countries [21, 22]. In an MCA, the chi-squared distance is calculated between the response patterns of all the individuals. The percentage of the total variation explained is calculated for each of the new dimensions, and the new dimensions are called components [23]. The coordinates of each country (the row profiles) in the lower dimensional space are extracted and used as a similarity score between country respondents [23]. We used similarity score estimates to construct an HCV policy implementation score. The similarity score was standardized to be out of 10 points using the standard min-max normalization method [24].

We developed two similarity scores: one based on WHO recommendation questions, received by all respondents, and another based on the verified policy questions, which were only sent to countries with the relevant policy in place. For the first similarity score for WHO recommendations, only a subset of the variables were included as we wanted to only incorporate variables that would be indicative of a country's commitment to HCV elimination as determined by multidisciplinary academic experts who were part of the study group. Fifteen experts were consulted by online questionnaire using SurveyMonkey. The survey used a 5-point Likert scale to evaluate which micro-elimination (targeted elimination that focuses on specific populations or geographies) and service integration variables the experts considered most important for inclusion into the score [25]. Questions that received a majority (greater than 50%) of positive responses were included into the score. The questionnaire is presented in appendix 2. The selected variables were then coded as 0 or 1 based on patient responses, with 0 representing a neutral, weak negative, or strong negative response, and 1 representing a slight positive or a strong positive response. We then applied the MCA to the coded data to generate the first similarity score.

The second similarity score was based on questions regarding verified policies. We used a subset of the questions about these policies, excluding questions with low numbers of respondents or low relevance, as determined during the expert consultation. The final verified questions were coded with three levels to capture the hierarchical nature of the question. The first level was coded as 0 and represented a country not receiving a question owing to not having that policy in place. The second level was coded as 1 and was assigned when a country had a verified policy in place and the patient group for that country responded with a weak negative or a strong negative response, indicating a lower level of implementation. The final level was coded as 2, which indicated that a country had a verified policy in place, and the patient respondent indicated a neutral, weak positive, or strong positive response to the implementation of said policy. In this analysis, Egypt, Israel, North Macedonia, Switzerland, and Turkey were excluded due to the lack of verified policies for opioid substitution therapy (OST) and needle syringe programmes (NSP) in the general population questions because the EMCDDA did not collect the information for these questions in these countries [20]. For the remaining countries, we then applied the MCA to the coded data to generate the second similarity score.

Proxy reference countries StagNation (no policies in place) and ElimiNation (all policies in place and implemented) were included as baseline in the first similarity score. In the second policy similarity score, ProcrastiNation (all policies, but none implemented) was also included so that the minimum and maximum of the similarity scores were representative of no implementation and maximum possible implementation respectively.

## Ethics

Since the study was approved and the data were stored in Denmark, we did not need ethics approval in any other country. According to the regional representative of the Danish data protection agency and the Barcelona Institute for Global Health (ISGlobal), this study was not considered human subjects research and therefore did not require ethical review or approval. We stored all raw data on secure servers in the Capital Region of Denmark and the data were managed according to Danish regulations. All off the code and data to reproduce this analysis can be found at https://osf.io/mh79q/.

## Results

Participants in 25 of 30 (83.3%) countries responded. Two of the countries that responded were from the Mediterranean Basin and 23 countries were from the EU/EEA or the UK.

The results of responses to the questions were considered for the policy similarity scores are presented in Table 2. Among WHO recommendation variables considered for inclusion, we limited the expert selection to only the micro-elimination and service integration subsets of questions (Table 2) as these questions fall along the same underlying concept. Experts indicated that for the micro-elimination variables, responses on migrants, PWID, prisoners, and sex workers were the most important for inclusion. For the service integration variables, the experts highlighted the importance of HCV service integration with blood safety, harm reduction, and migrant health services. We used the binary responses to these seven items to generate the first similarity score for the countries.

In the MCA for WHO recommendations, the first component accounted for 75.5% of the total inertia in the subset data, whereas the second component accounted for 2.9% of inertia (S1 Fig). For the first component, all of the negative outcomes of the dichotomous variables were associated with negative scores, while all of the positive outcomes of the dichotomous variables were associated with positive scores (S1 Fig). The variables with the greatest weight in determining a positive score were a positive response towards HCV micro-elimination among migrants, prisoners and sex workers and the integration of HCV testing and treatment with migrant health services. Conversely, the responses that were associated with the most negative scores were not having HCV micro-elimination efforts among PWID, not having HCV integration with harm reduction services, and a lack of HCV blood safety testing (S1 Fig).

The extracted row profiles for the countries from WHO recommendation MCA are presented in Fig 1. The countries with the highest scores were Cyprus, Netherlands, Portugal, Slovenia, and Sweden (all having similarity scores of 10). These countries responded positively to all of the binary indicators in WHO recommendation similarity score. The lowest scoring country was Bulgaria (score of 0), which responded negatively to all but two of the binary indicators. All of the other scores ranged between 0 and 10, forming a distribution of countries along the spectrum of policy elimination (Fig 1).

For the verified policies similarity score, we considered for inclusion all the questions under the subheading "verified questions" in Table 2. We included the following variables: fibrosis HCV treatment restrictions; NSP in the general population; OST in the general population; NSP in prisons; OST in prisons; testing/screening for HCV in prisons; and the treatment of HCV in prisons. The MCA of these variables yielded a two-dimensional solution in which the first component accounted for 44.3% of the inertia in the data, and the second component accounted for 34.6% of the inertia in the dataset (S2 Fig). The responses that mapped the closest to the first component were whether a country had a verified policy in place or not, with positive values indicating the presence of verified policies and negative values indicating the absence of verified policies. The second dimension was interpreted as whether a policy was being implemented in practice. Negative values represent policies that are in place but not well implemented, and positive values represent policies that are in place and well implemented (S2 Fig). The coordinates for both of the dimensions of the second policy similarity score were extracted and are presented in Fig 2.

Spain had the highest first component score (10) of the verified policies similarity score (i.e. the presence or absence of verified policies), followed by Portugal (9.48), Sweden (9.48), Austria (9.48), and the United Kingdom (9.48). Bosnia and Herzegovina, which only had an OST and NSP programme for non-incarcerated individuals, had the lowest score 3.87 (Fig 2). Cyprus had the next lowest score, which also only had an OST and NSP programme, like Ukraine, but were reported as being well implemented.

Spain had the highest score of the second component as well (the proportion of verified policies that are well implemented, according to the patient group respondent) with a maximum score of 10 (Fig 2). The lowest score was Finland, which had a score of 5.54. The patient respondent reported that of all the policies in place, NSP in the general population, OST in the

**Table 2. Likert point breakdown of questions and sub-questions considered for inclusion into the policy similarity scores.**

| | Strong positive | Weak positive | Neutral | Weak negative | Strong negative | Not applicable |
|---|---|---|---|---|---|---|
| **Total responses, n** | 243 | 192 | 166 | 126 | 89 | 134 |
| *WHO recommendation questions, n (%)* | | | | | | |
| **GHSS monitoring goal** | 3 (1.2) | 8 (4.2) | 6 (3.6) | 5 (4.0) | 3 (3.4) | 0 (0.0) |
| **Policy barriers equity GHSS** | 5 (2.1) | 8 (4.2) | 8 (4.8) | 3 (2.4) | 1 (1.1) | 0 (0.0) |
| **Policy stigma and discrimination** | 0 (0.0) | 8 (4.2) | 9 (5.4) | 5 (4.0) | 3 (3.4) | 0 (0.0) |
| *WHO recommendation: Micro elimination sub-questions, n (%)* | | | | | | |
| **Generational cohort** | 2 (0.8) | 3 (1.6) | 11 (6.6) | 4 (3.2) | 5 (5.6) | 0 (0.0) |
| **Haemodialysis patients** | 16 (6.6) | 3 (1.6) | 4 (2.4) | 2 (1.6) | 0 (0.0) | 0 (0.0) |
| **Haemophilia patients** | 16 (6.6) | 3 (1.6) | 4 (2.4) | 2 (1.6) | 0 (0.0) | 0 (0.0) |
| **Men who have sex with men** | 5 (2.1) | 9 (4.7) | 6 (3.6) | 3 (2.4) | 2 (2.2) | 0 (0.0) |
| **Migrants*** | 2 (0.8) | 3 (1.6) | 6 (3.6) | 7 (5.6) | 7 (7.9) | 0 (0.0) |
| **Patients with advanced liver disease** | 12 (4.9) | 9 (4.7) | 2 (1.2) | 2 (1.6) | 0 (0.0) | 0 (0.0) |
| **People living with HIV** | 17 (7.0) | 6 (3.1) | 1 (0.6) | 1 (0.8) | 0 (0.0) | 0 (0.0) |
| **People who inject drugs*** | 9 (3.7) | 9 (4.7) | 4 (2.4) | 3 (2.4) | 0 (0.0) | 0 (0.0) |
| **Prisoners*** | 6 (2.5) | 3 (1.6) | 7 (4.2) | 7 (5.6) | 2 (2.2) | 0 (0.0) |
| **Sex workers*** | 2 (0.8) | 5 (2.6) | 4 (2.4) | 8 (6.3) | 6 (6.7) | 0 (0.0) |
| **Thalassaemia patients** | 9 (3.7) | 1 (0.5) | 5 (3.0) | 3 (2.4) | 7 (7.9) | 0 (0.0) |
| **Transgender people** | 1 (0.4) | 1 (0.5) | 7 (4.2) | 9 (7.1) | 7 (7.9) | 0 (0.0) |
| **Transplantation patients** | 17 (7.0) | 5 (2.6) | 1 (0.6) | 0 (0.0) | 2 (2.2) | 0 (0.0) |
| **Veteran/military personnel** | 6 (2.5) | 3 (1.6) | 5 (3.0) | 3 (2.4) | 8 (9.0) | 0 (0.0) |
| *WHO recommendation: Service integration sub-questions, n (%)* | | | | | | |
| **Alcohol use services** | 2 (0.8) | 5 (2.6) | 5 (3.0) | 7 (5.6) | 6 (6.7) | 0 (0.0) |
| **Blood safety*** | 17 (7.0) | 4 (2.1) | 1 (0.6) | 3 (2.4) | 0 (0.0) | 0 (0.0) |
| **Cancer prevention and management** | 3 (1.2) | 8 (4.2) | 7 (4.2) | 5 (4.0) | 2 (2.2) | 0 (0.0) |
| **Haemodialysis centres** | 16 (6.6) | 4 (2.1) | 3 (1.8) | 2 (1.6) | 0 (0.0) | 0 (0.0) |
| **Harm reduction services*** | 5 (2.1) | 10 (5.2) | 5 (3.0) | 3 (2.4) | 2 (2.2) | 0 (0.0) |
| **HIV treatment clinic** | 15 (6.2) | 7 (3.6) | 2 (1.2) | 1 (0.8) | 0 (0.0) | 0 (0.0) |
| **Migrant health services*** | 3 (1.2) | 3 (1.6) | 4 (2.4) | 7 (5.6) | 8 (9.0) | 0 (0.0) |
| **NCD prevention and management** | 2 (0.8) | 4 (2.1) | 7 (4.2) | 6 (4.8) | 6 (6.7) | 0 (0.0) |
| **STI and reproductive health clinics** | 5 (2.1) | 11 (5.7) | 5 (3.0) | 2 (1.6) | 2 (2.2) | 0 (0.0) |
| *Verified questions, n (%)* | | | | | | |
| **Drug and alcohol restriction**** | 5 (2.1) | 6 (3.1) | 4 (2.4) | 2 (1.6) | 1 (1.1) | 7 (5.2) |
| **Non-hospital testing** | 3 (1.2) | 3 (1.6) | 3 (1.8) | 2 (1.6) | 1 (1.1) | 13 (9.7) |
| **Non-hospital treatment** | 0 (0.0) | 5 (2.6) | 0 (0.0) | 0 (0.0) | 0 (0.0) | 20 (14.9) |
| **Non-specialist treatment** | 0 (0.0) | 0 (0.0) | 1 (0.6) | 0 (0.0) | 0 (0.0) | 24 (17.9) |
| **NSP general population**** | 5 (2.1) | 5 (2.6) | 6 (3.6) | 4 (3.2) | 1 (1.1) | 4 (3.0) |
| **NSP prison**** | 0 (0.0) | 0 (0.0) | 2 (1.2) | 1 (0.8) | 0 (0.0) | 22 (16.4) |
| **OST general population**** | 8 (3.3) | 6 (3.1) | 5 (3.0) | 2 (1.6) | 0 (0.0) | 4 (3.0) |
| **OST prison**** | 4 (1.6) | 5 (2.6) | 4 (2.4) | 2 (1.6) | 1 (1.1) | 9 (6.7) |
| **Testing and screening of HCV in prisons**** | 2 (0.8) | 5 (2.6) | 4 (2.4) | 3 (2.4) | 2 (2.2) | 9 (6.7) |
| **Treatment of HCV in prisons**** | 3 (1.2) | 6 (3.1) | 3 (1.8) | 3 (2.4) | 1 (1.1) | 9 (6.7) |
| **Fibrosis treatment restrictions for HCV**** | 7 (2.9) | 3 (1.6) | 1 (0.6) | 1 (0.8) | 0 (0.0) | 13 (9.7) |

Variables denoted with * were included into the first similarity score based off WHO recommendations and variables that are marked with ** were included into the second similarity score based off verified policies. GHSS = Global Health Sector Strategy, NCD = non-communicable disease, STI = sexually transmitted infection, NSP = needle-syringe programme, OST = opioid substitution therapy, HCV = hepatitis C virus

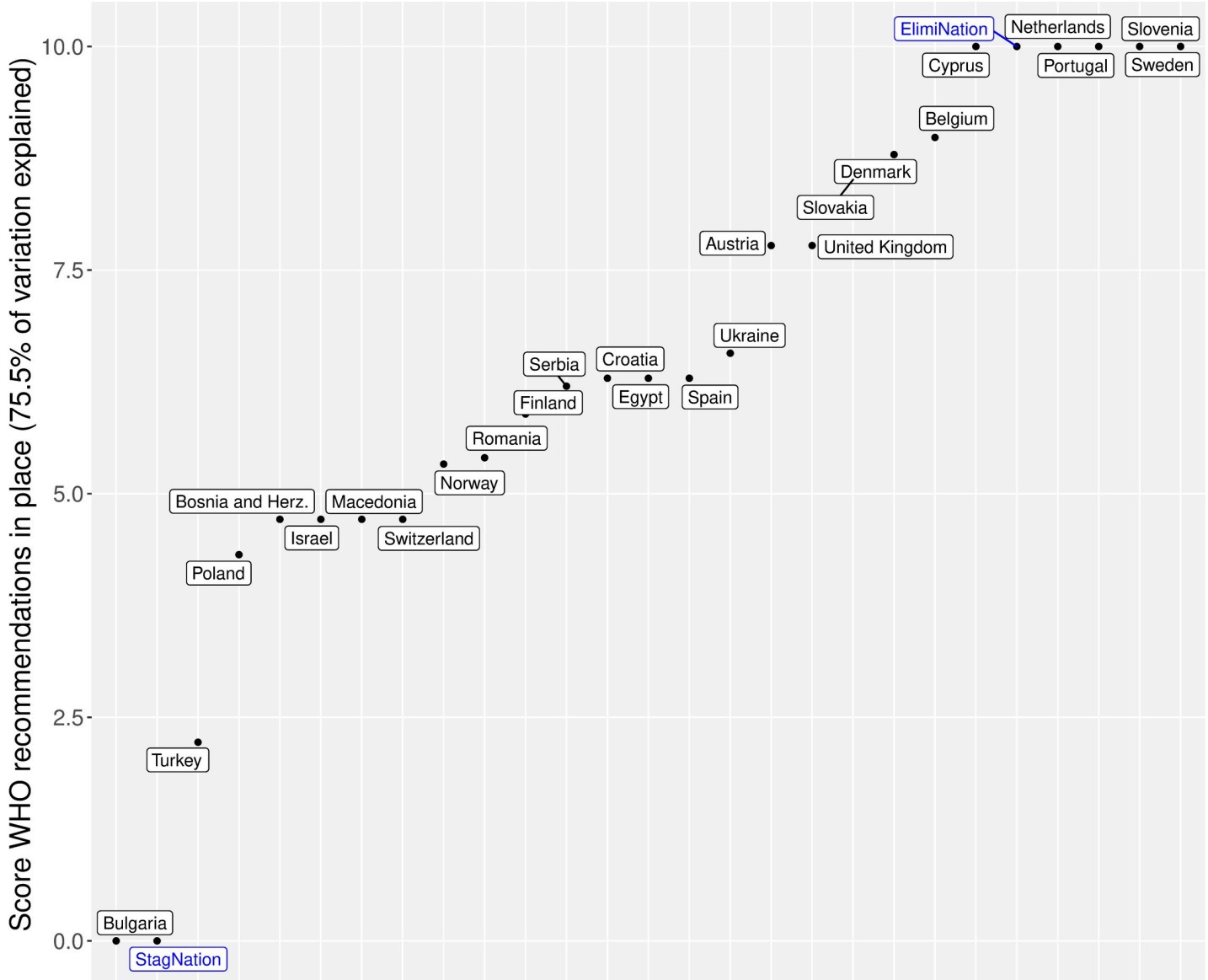

**Fig 1. WHO recommendation similarity scores for the studied countries.** Summary of the standardized scores for the different countries according to row profiles of the first components of the MCA.

general population, and HCV testing/screening in prisons were being adequately implemented. However, drug–alcohol restrictions, the lifting of fibrosis severity HCV treatment restrictions, and HCV treatment in prisons were not well implemented.

Table 3 presents the reported similarity score for WHO recommendations, the two similarity scores for the verified policies, the difference between the WHO recommendation similarity score and the first verified policy similarity score for each applicable country, and the mean confidence score of the respondent for each country. Positive values of the difference between WHO recommendation similarity score and the first similarity score for the verified policies may reflect poor engagement with patients in the discussion of the current status of HCV elimination in their country. Negative values of this difference could suggest poor implementation of policies. The country with the largest negative gap between the verified policy similarity

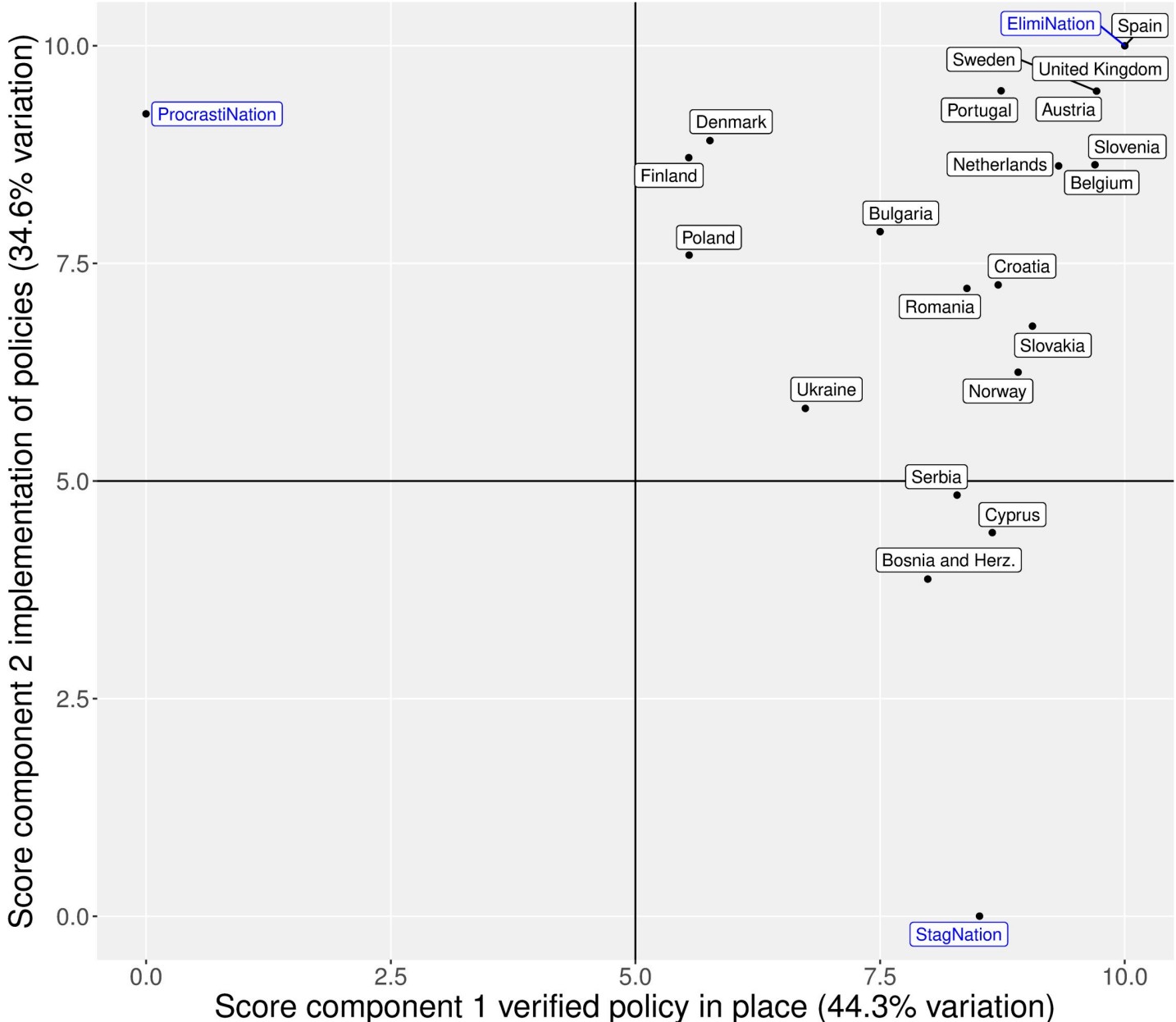

**Fig 2. Verified policy similarity scores for the different countries (n = 20).** The similarity score of the countries based on their responses to the verified questions and whether a particular policy was in place.

score and WHO recommendation score was Bulgaria, with a difference of –7.86. The largest positive gap was Cyprus, with a difference of 5.59 between WHO recommendation score and the first dimension of the verified policy similarity score. The mean confidence score for Bulgaria was 2.53, and for Cyprus the mean confidence score was 3.38 (Table 3).

## Discussion

The Hep-CORE 2018 study focused on how well countries were implementing WHO recommendations from the GHSS as well as verified policies to eliminate viral hepatitis. We analyzed

**Table 3. The WHO recommendation similarity score of the two similarity scores from the verified policies, the difference between WHO recommendation score and first similarity score of verified policies, and the mean confidence score for all of the countries.**

| Country | WHO recommendation similarity score | Similarity score verified policies dimension 1 | Difference of WHO score and verified policy dim 1 | Similarity score verified policies dimension 2 | Overall confidence mean (SD) |
|---|---|---|---|---|---|
| Bulgaria | 0 | 7.86 | -7.86 | 7.5 | 2.53 (0.77) |
| Spain | 6.29 | 10 | -3.71 | 10 | 2.65 (0.61) |
| Poland | 4.32 | 7.59 | -3.27 | 5.55 | 3.24 (0.75) |
| Finland | 5.89 | 8.71 | -2.82 | 5.54 | 3.11 (0.46) |
| Romania | 5.4 | 7.21 | -1.81 | 8.39 | 3.29 (0.69) |
| United Kingdom | 7.77 | 9.48 | -1.71 | 9.71 | 3.70 (0.66) |
| Austria | 7.77 | 9.48 | -1.71 | 9.71 | 3.06 (0.64) |
| Croatia | 6.29 | 7.25 | -0.96 | 8.71 | 3.56 (0.51) |
| Norway | 5.33 | 6.25 | -0.92 | 8.91 | 3.21 (0.89) |
| Denmark | 8.79 | 8.91 | -0.12 | 5.76 | 1.95 (0.52) |
| Belgium | 8.98 | 8.63 | 0.35 | 9.7 | 2.12 (0.81) |
| Sweden | 10 | 9.48 | 0.52 | 9.71 | 3.11 (0.83) |
| Portugal | 10 | 9.48 | 0.52 | 8.74 | 3.89 (0.32) |
| Ukraine | 6.57 | 5.83 | 0.74 | 6.74 | 2.71 (0.47) |
| Bosnia and Herzegovina | 4.71 | 3.87 | 0.84 | 7.99 | 2.69 (0.48) |
| Serbia | 6.2 | 4.84 | 1.36 | 8.29 | 3.56 (0.63) |
| Slovenia | 10 | 8.63 | 1.37 | 9.7 | 3.76 (0.44) |
| Netherlands | 10 | 8.62 | 1.38 | 9.32 | 2.67 (0.97) |
| Slovakia | 8.51 | 6.78 | 1.73 | 9.06 | 3.19 (0.40) |
| Cyprus | 10 | 4.41 | 5.59 | 8.65 | 3.38 (0.51) |
| Switzerland | 4.71 | NA | NA | NA | 3.77 (0.44) |
| Egypt | 6.29 | NA | NA | NA | 3.90 (0.32) |
| Turkey | 2.22 | NA | NA | NA | 3.83 (0.39) |
| North Macedonia | 4.71 | NA | NA | NA | 3.42 (0.52) |
| Israel | 4.71 | NA | NA | NA | 4.00 (0) |
| ProcrastiNation | NA | 9.22 | NA | 0 | NA |
| ElimiNation | 10 | 10 | - | 10 | NA |
| StagNation | 0 | 0 | - | 8.52 | NA |

the similarity of different country responses for a subset of WHO recommendations and a subset of verified policy questions. Patient group respondents in 25 countries indicated incomplete implementation of WHO recommended HCV policy implementation across Europe, with concerning intercountry discrepancies evidenced.

Currently, only 12 countries are reported to be on track to meet WHO elimination targets for HCV—eight of these are in WHO European Region and seven were among our study countries [19]. Countries that are on track and included in this study are Egypt, the Netherlands, Spain, Switzerland, and the UK. France and Italy are both on track and patients' groups were sent surveys but did not respond. For Egypt and Switzerland, we did not calculate a difference because both countries were excluded from the verified similarity analysis and therefore did not have a score. The difference for both the UK and Spain were negative, suggesting the under-implementation of policies from the perspective of the respondents. In contrast, the Netherlands had a positive difference, suggesting that policy-makers and the medical community could better engage patient advocacy groups to highlight policies that are in place.

Although Egypt and Switzerland did not have a score generated from the verified similarity score, both had low scores (6.29 and 4.71, respectively) relative to the possible maximum of 10

for WHO recommendation similarity score. These low scores indicate that vulnerable groups, such as migrants, are not targeted well. This finding highlights an issue with WHO 2030 goals and what is considered the "elimination" of HCV as a public health threat. For example, there is a good probability that both Switzerland and Egypt will meet the WHO 2030 HCV elimination targets at a population level despite not targeting all vulnerable groups. As a consequence, some vulnerable groups may be left behind. Therefore, it is important to assess if and how the most vulnerable and/or marginalized patients are included in policy and implementation of governmental HCV elimination strategies, even if a country might be on track for HCV elimination. This idea is further reinforced by the findings of the recent Lancet Commission on the acceleration of viral hepatitis elimination [14]. In the Commission, a policy score card was created for select countries, including five (Egypt, Poland, Romania, Spain, and the UK) that overlap with our study. The scores we generated indicated that Egypt could be leaving marginalized populations, like PWID, behind. Likewise, the Lancet Commission, found that Egypt does not have a harm reduction policy in place. The Lancet Commission also identified policy gaps in Poland and Romania, such as the lack of a publicly-funded screening programme and reliable national epidemiological data, which supports the low scores we recorded for the countries in this study. Conversely, the Commission reported that Spain and the UK have both of these policies in place, and we found that these countries scored highly, outranking Egypt, Poland, and Romania [14].

## Limitations

The Hep-CORE 2018 study relies on only one stakeholder group for information; therefore, inferential statistics cannot be conducted at the country level as we only have a single replicate. Additionally, this means that a random sample was not possible as we could not randomly select from any groups. Although, a random sample for this kind of research is not desirable since we surveyed key stakeholders whose perspectives were most relevant to the research questions rather than those of random patient groups. We understand that our sampling method could introduce a bias since respondents likely had similar education and sociodemographic characteristics. However, this selection is desirable in our study as we tried to obtain an accurate view of HCV policy implementation from patient groups rather than determine patients' knowledge about HCV policy. Furthermore, the survey could not be altered once it was launched, potentially leading to outcome censoring if a policy changed while the survey was being completed. For example, this survey was conducted in Q4 2018, but Egypt started a national screening programme that would include all subgroups in the study in Q2 2019, which would not be captured by this study. Furthermore, Egypt's new approach is representative of where a macro-elimination approach is possible due to government commitment, and micro-elimination in this scenario is not necessarily needed [26]. Another limitation of the cross-sectional nature of this study is that a policy might not have been in place long enough for a patient group to accurately evaluate its implementation. In the future, it would be ideal if the data needed to build the score could be collected every 6 months and serve as a progress monitoring tool; however, a lack of funding constrains the ability to create such projects and efforts. Additionally, involving drug user groups and other organisations whose members might not be well represented by patient groups would strengthen reporting. Additionally, the survey was only disseminated in English, which may have limited the accuracy of the responses as most respondents were non-native English speakers.

For future studies, it will be important to explore new approaches to hepatitis policy monitoring, such as cross-referencing data collected from civil society with the data from other sources including publicly available DAA reimbursement records, government health officials,

and clinicians. The generalizability of the MCA method used to evaluate the responding countries is another important limitation. MCA methods used to make the similarity scores tend to produce unstable results because they depend on the sample used. The inclusion of anomalous data, choices carried out by the researchers when reducing the categories of the variables, and other factors could have affected the final results and country rankings [27]. However, the use of standardized countries helped stabilize those estimates and provide meaning to the numerical values of the similarity score, a technique we have validated elsewhere [28]. Although, that study does not levy patient or civil society information like this present study. Finally, noting that the scores are derived from patient responses, these policy similarity scores should not be used in isolation but as an aid in decisionmaking [29].

## Conclusions

Patient groups from all 25 studied countries identified areas of improvement in HCV policy development and implementation with substantial inter-country variation. Policy improvements are essential for the entire European Region to achieve the WHO 2030 viral hepatitis elimination goal. Civil society and the government must take more action to monitor implementation of current policies. Crucially, European countries must ensure acceptable and accessible HCV service provision for high-risk populations, who continue to be overlooked in many settings.

## Supporting information

**S1 Fig. MCA outputs for WHO recommendation index.** The MCA determined weight for all of the categories for the WHO recommendation index.
(TIF)

**S2 Fig. MAC outputs for the verified index.** The MCA determined weights for all of the categories for the verified index.
(TIF)

**S1 File. Hep-CORE survey.** This is the complete survey that was used along with code fields.
(PDF)

**S2 File. Weighting expert survey.** This is the complete survey that was used to ask experts what should be included into the index.
(PDF)

## Acknowledgments

We would like to thank all of the participants in this study, the Hep-CORE study group (led by JVL–email: Jeffrey.lazarus@isglobal.org –in addition to the authors of this paper they include Charles Gore, Greet Hendrickx, Achim Kautz, Luis Mendao, Antons Mozalevskis, Tatjana Reij, and Eberhard Schatz. We would also like to thank ELPA for their collaboration throughout the study.

## Author Contributions

**Conceptualization:** Jeffrey V. Lazarus.

**Data curation:** Adam Palayew, Samya R. Stumo.

**Formal analysis:** Adam Palayew, Graham S. Cooke, Sharon J. Hutchinson, Homie Razavi.

**Funding acquisition:** Jeffrey V. Lazarus.

**Methodology:** Adam Palayew, Samya R. Stumo, Sharon J. Hutchinson.

**Project administration:** Adam Palayew, Jeffrey V. Lazarus.

**Software:** Adam Palayew.

**Supervision:** Jeffrey V. Lazarus.

**Validation:** Adam Palayew, Samya R. Stumo.

**Visualization:** Adam Palayew.

**Writing – original draft:** Adam Palayew, Samya R. Stumo, Jeffrey V. Lazarus.

**Writing – review & editing:** Adam Palayew, Graham S. Cooke, Sharon J. Hutchinson, Marie Jauffret-Roustide, Mojca Maticic, Magdalena Harris, Ammal M. Metwally, Homie Razavi, Jeffrey V. Lazarus.

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
