## [Decision Letter · Decision Letter 0]

1 Jun 2020

PONE-D-20-09886

The Hep-CORE Policy Score: A European hepatitis C national policy implementation ranking

PLOS ONE

Dear Dr. Lazarus,

Thank you for submitting your manuscript to PLOS ONE. After careful consideration, we feel that it has merit but does not fully meet PLOS ONE’s publication criteria as it currently stands. Therefore, we invite you to submit a revised version of the manuscript that addresses the points raised during the review process.

I enjoyed the study results however there are concerns from reviewers should be considered carefully before acceptance of manuscript for publication in PLOS ONE.

We look forward to receiving your revised manuscript.

Kind regards,

Heidar Sharafi

Academic Editor

PLOS ONE

Additional Editor Comments:

1. I tried to access the data of the study however it needed the approval from authors and also after being approved nothing was found in the folders. Authors should follow the data policy of PLOS.

2. I could not find the supplementary Figures 1 and 2 in the submission.

'The European Liver Patients’ Association received funding from AbbVie, Gilead Sciences and MSD to carry out this study. The funders did not have any control over the study or the resulting manuscript. ICMJE forms have been provided by all authors.'

We note that you received funding from a commercial source: AbbVie, Gilead Sciences and Merck & Co.

Pb. lease include your amended Competing Interests Statement within your cover letter. We will change the online submission form on your behalf.

'GC is supported in part by the BRC of Imperial College NHS Trust and an NIHR Professorship. JVL is supported by a Spanish Ministry of Science, Innovation and Universities Miguel Servet grant (Instituto de Salud Carlos III/ESF, European Union (CP18/00074)). JVL further acknowledges support from the Spanish Ministry of Science, Innovation and Universities through the “Centro de Excelencia Severo Ochoa 2019-2023” Program (CEX2018-000806-S), and support from the Government of Catalonia through the CERCA Program.'

'The European Liver Patients’ Association received funding from AbbVie, Gilead Sciences and MSD to carry out this study. The funders did not have any control over the study or the resulting manuscript. ICMJE forms have been provided by all authors.'

4. One of the noted authors is a group; the Hep-CORE Study Group.

In addition to naming the author group, please list the individual authors and affiliations within this group in the acknowledgments section of your manuscript.

Please also indicate clearly a lead author for this group along with a contact email address.

5. Please upload a copy of Figure 5, to which you refer in your text on page 14. If the figure is no longer to be included as part of the submission please remove all reference to it within the text.

6. Please include captions for your Supporting Information files at the end of your manuscript, and update any in-text citations to match accordingly. Please see our Supporting Information guidelines for more information: http://journals.plos.org/plosone/s/supporting-information

Reviewers' comments:

Reviewer's Responses to Questions

**Comments to the Author**

1. Is the manuscript technically sound, and do the data support the conclusions?

Reviewer #1: Yes

Reviewer #2: Yes

Reviewer #3: Partly

2. Has the statistical analysis been performed appropriately and rigorously? 

Reviewer #1: I Don't Know

Reviewer #2: Yes

Reviewer #3: I Don't Know

3. Have the authors made all data underlying the findings in their manuscript fully available?

Reviewer #1: Yes

Reviewer #2: Yes

Reviewer #3: Yes

4. Is the manuscript presented in an intelligible fashion and written in standard English?

Reviewer #1: No

Reviewer #2: Yes

Reviewer #3: Yes

5. Review Comments to the Author

Reviewer #1: Dear associate editor of PLOS ONE

Many thanks for your kind invitation to review the manuscript entitled “The Hep-CORE Policy Score: A European Hepatitis C national policy implementation ranking”

In this project, 25 liver patient organizations representing 25 countries in Europe have been evaluated regarding policies for HCV elimination. Two sets of surveys have been performed (based on WHO policies and based on sources verifying existing policy). Then they determined scores for each county and finally they concluded that European region is not following WHO policies for HCV elimination goals. My comments:

1. Introduction section is well written. However, I think that authors should add a section to it regarding the aims of their project.

2. I think that the method of the abstract section is a little complicated and should be presented more simple than this so that all readers can fully understand it.

3. Regarding the methods of this project, I suggest to use an expert to review. I think it is too complicated to for a usual reader to understand it.

4. Items of questioners have been explained in detail. Sampling methods have been well discussed. I also think that this study has not ethical problems.

5. I suggest that authors summarize the results section. In this section readers encounter with huge data and it leads to confusion. I think some of them should be transferred in the supplementary section. I recommend that authors use some figures for better representing their data.

Overall comment:

The results of this study seems to be very important and it seriously needs attention. However, I think that the manuscript should be summarized and written a in better style for better understanding.

Best

reviewer

Reviewer #2: Dear Editor

Thanks for inviting me to review the manuscript PONE-D-20-09886 entitled "The Hep-CORE Policy Score: A European hepatitis C national policy implementation ranking". I have only two minor comments to the authors:

- The scoring method for each item may be unclear for the readers. It could be better if you mention which score belongs to which answer. Also, please mention if all items had similar scores for Strong positive, Weak positive, Neutral, Weak negative,

and Strong negative answers. If not, please specify which one has a different scoring.

- Please consider two decimals for each number showing a central tendency or SD in table 3 (e.g. 5.40 instead of 5.4).

Regards

Reviewer

Reviewer #3: This study assess the extent of policy implementation from the perspective of liver patient groups. However the implications of patients groups seems minimal, only answering the survey. None of these organizations are among the authors.

The study have some important limitations. Firstly would be important to have more details on the patients organizations included in the study. They represented the country or are more local associations? To translate the answers of a patient’s organization to all country could not be representative of the country. Secondly, the survey was performed in 2018 and the field is moving quickly, the authors should comment on that

Two of the countries that responded were from the Mediterranean Basin and 23 countries were from the EU/EEA or the UK. Are these two countries outside EU/EEA?

134 surveys had missed responses? The impact on the missing response and the countries with missed responses should be discuss and give more details (table2)

Please comment that two big European countries such as Italy and Germany are not included

The title of the manuscript is not accurate. The authors have to add patients organizations in the title because the manuscript reflects the point of view of these organizations

6. PLOS authors have the option to publish the peer review history of their article (what does this mean?). If published, this will include your full peer review and any attached files.

Reviewer #1: No

Reviewer #2: Yes: Hamidreza Karimi-Sari, MD

Reviewer #3: No

---

## [Author Response · Author response to Decision Letter 0]

19 Jun 2020

Additional Editor Comments:

  1. I tried to access the data of the study however it needed the approval from authors and also after being approved nothing was found in the folders. Authors should follow the data policy of PLOS.

Response: Full access has been given for the folders and the data files and coding scripts should be fully accessible. Please contact me if you cannot access the files.

2. I could not find the supplementary Figures 1 and 2 in the submission.

Response: Supplementary figures have been provided as well as supplementary files with the naming format of PLOS ONE

Journal requirements: When submitting your revision, we need you to address these additional requirements.

Response: We have formatted our manuscript and file names according to the guidelines in the link above.

'The European Liver Patients’ Association received funding from AbbVie, Gilead Sciences and MSD to carry out this study. The funders did not have any control over the study or the resulting manuscript. ICMJE forms have been provided by all authors.'We note that you received funding from a commercial source: AbbVie, Gilead Sciences and Merck & Co.

Response: We have amended our competing interest statement in our cover letter as requested. Please let us know if you require more information. 

'GC is supported in part by the BRC of Imperial College NHS Trust and an NIHR Professorship. JVL is supported by a Spanish Ministry of Science, Innovation and Universities Miguel Servet grant (Instituto de Salud Carlos III/ESF, European Union (CP18/00074)). JVL further acknowledges support from the Spanish Ministry of Science, Innovation and Universities through the “Centro de Excelencia Severo Ochoa 2019-2023” Program (CEX2018-000806-S), and support from the Government of Catalonia through the CERCA Program.'

'The European Liver Patients’ Association received funding from AbbVie, Gilead Sciences and MSD to carry out this study. The funders did not have any control over the study or the resulting manuscript. ICMJE forms have been provided by all authors.'

Response: We have fixed our acknowledgement and funding section and have added the amended funding section to the cover letter. Please let us know if you require further information. 

4. One of the noted authors is a group; the Hep-CORE Study Group.

In addition to naming the author group, please list the individual authors and affiliations within this group in the acknowledgments section of your manuscript.

Response: We have added to acknowledgements that the lead of the Hep-CORE study group is JVL who is also the corresponding author for the paper and includes GSC, SJH, MJR, MM, MH, AMM, and HR among the authors and Charles Gore, Greet Hendrickx, Achim Kautz, Luis Mendao, Antons Mozalevskis, Tatjana Reic, and Eberhard Schatz who were not among the authors. 

.Please also indicate clearly a lead author for this group along with a contact email address.

Response: We have not added the email but indicate that the lead of the study group is also the corresponding author in the acknowledgments of the paper and thus can be contacted.

5. Please upload a copy of Figure 5, to which you refer in your text on page 14. If the figure is no longer to be included as part of the submission please remove all reference to it within the text.

Response: Figure 5 was a typo and it meant to read figure 2. We have corrected this.

6. Please include captions for your Supporting Information files at the end of your manuscript, and update any in-text citations to match accordingly. Please see our Supporting Information guidelines for more information: http://journals.plos.org/plosone/s/supporting-information

Response: We have made a Supporting information section with the names and captions 

  Reviewers' comments:  

Reviewer's Responses to Questions 

  Reviewer #1: Dear associate editor of PLOS ONE Many thanks for your kind invitation to review the manuscript entitled “The Hep-CORE Policy Score: A European Hepatitis C national policy implementation ranking” In this project, 25 liver patient organizations representing 25 countries in Europe have been evaluated regarding policies for HCV elimination. Two sets of surveys have been performed (based on WHO policies and based on sources verifying existing policy). Then they determined scores for each county and finally they concluded that European region is not following WHO policies for HCV elimination goals

My comments: 1. Introduction section is well written. However, I think that authors should add a section to it regarding the aims of their project.

Response: We believe that this requirement is satisfied on page 4 on lines 109 to 111 with the text, “Inspired by this call, Hep-CORE 2018 set out to assess whether known national HCV policies in Europe, the United Kingdom (UK), and the Mediterranean Basin are functioning in practice and to score the extent of implementation according to patient advocacy groups” 

 2. I think that the method of the abstract section is a little complicated and should be presented more simple than this so that all readers can fully understand it.

Response: We have tried to simplify the methods in the abstract to make them simpler and straight forward and avoid a lot of potentially misleading jargon. 

 3. Regarding the methods of this project, I suggest to use an expert to review. I think it is too complicated to for a usual reader to understand it.

Response: We agree that the methods in this paper can be a little daunting, but for what it is worth we consulted one of the world expert on the methodology who wrote the text book on the subject, Dr Michael Greenacre. Based on his input, we developed methods.

 4. Items of questioners have been explained in detail. Sampling methods have been well discussed. I also think that this study has not ethical problems.

Response: Thank you.

 5. I suggest that authors summarize the results section. In this section readers encounter with huge data and it leads to confusion. I think some of them should be transferred in the supplementary section. I recommend that authors use some figures for better representing their data.

Response: We have tried to simplify the result; however, there is a lot of information in this study to present and some of this is unavoidable. We believe that we have presented this large volume of information in a logical fashion. 

 

Overall comment: The results of this study seems to be very important and it seriously needs attention. However, I think that the manuscript should be summarized and written a in better style for better understanding. Best reviewer  Reviewer #2: Dear Editor Thanks for inviting me to review the manuscript PONE-D-20-09886 entitled "The Hep-CORE Policy Score: A European hepatitis C national policy implementation ranking". I have only two minor comments to the authors:  - The scoring method for each item may be unclear for the readers. It could be better if you mention which score belongs to which answer. Also, please mention if all items had similar scores for Strong positive, Weak positive, Neutral, Weak negative, and Strong negative answers. If not, please specify which one has a different scoring.

Response: This information is provided in the supplementary figures that shows the value of how much each level of response was worth. The reviewer; however, must be mistaken on the value we assigned as we did not assign values to each of the 5 levels but rather we grouped the variables and analyzed them as three distinct levels of negative, neutral, and positive, which have weights in the scores for each item. These weights were the ones used in the score and are part of the supplemental figures.  - Please consider two decimals for each number showing a central tendency or SD in table 3 (e.g. 5.40 instead of 5.4).

Response: This would not be appropriate as for simple count data with values less than 10 and the n being in the double digits the correct number of decimals would be one. We maintain that we should only present to one decimal and that the data provided do not allow us to present two decimals even though we could output it this would not be scientifically accurate due to the rules of significant digits. 

 Regards Reviewer  Reviewer #3: This study assess the extent of policy implementation from the perspective of liver patient groups. However the implications of patients groups seems minimal, only answering the survey. None of these organizations are among the authors.

Response: We disagree as both Dr Magdalena Harris and Dr Ammal M Metwally are both patients with lived experience as well as academics and are authors of the paper. 

 The study have some important limitations. Firstly would be important to have more details on the patients organizations included in the study. They represented the country or are more local associations? To translate the answers of a patient’s organization to all country could not be representative of the country. Secondly, the survey was performed in 2018 and the field is moving quickly, the authors should comment on that

Response: We have provided information on the patient organizations already in the paper and do not feel like we can expand in part because there are aspects that we cannot include about the patient organization due to confidentiality reasons such as who answered the survey. The fact that the survey was only in English is now mentioned in the limitation on Page 18 lines 398 to 400. We have also already discussed the cross-sectional nature of the study in the limitations and re the need to comment on the survey being done in 2018, please page 18 lines 387 to 391 as well as lines 393 to 397

  Two of the countries that responded were from the Mediterranean Basin and 23 countries were from the EU/EEA or the UK. Are these two countries outside EU/EEA?

Response: Yes they are outside the EU/EEA and neither are the UK.

 134 surveys had missed responses? The impact on the missing response and the countries with missed responses should be discuss and give more details (table2)

Response: I understand why the reviewer may be confused, but we had complete data for all the responses so we did not have missing data besides the non-responses which were for 3 entire countries and is a refusal to participate not missing data per say. To address the concern of the reviewers we have removed the “/missing” from the header in Table 2 so that it only reads non-applicable which is the correct designation of everything in that column.

 Please comment that two big European countries such as Italy and Germany are not included The title of the manuscript is not accurate. The authors have to add patients organizations in the title because the manuscript reflects the point of view of these organizations

Response: We have added patient organizations in the title to make it more informative, “The Hep-CORE Policy Score: A European hepatitis C national policy implementation ranking with data from patient organizations”. We cannot elaborate further on why Italy and Germany were not included as the patient organizations from those countries declined to participate.   

Response: All of our figures have been put through PACE and are to the standards of PlosONE

---

## [Editor Report · Decision Letter 1]

22 Jun 2020

The Hep-CORE Policy Score: A European hepatitis C national policy implementation ranking based on patient organization data

PONE-D-20-09886R1

Dear Dr. Lazarus,

We’re pleased to inform you that your manuscript has been judged scientifically suitable for publication and will be formally accepted for publication once it meets all outstanding technical requirements.

Kind regards,

Heidar Sharafi

Academic Editor

PLOS ONE

---

## [Editor Report · Acceptance letter]

8 Jul 2020

PONE-D-20-09886R1 

The Hep-CORE Policy Score: A European hepatitis C national policy implementation ranking based on patient organization data 

Dear Dr. Lazarus:

I'm pleased to inform you that your manuscript has been deemed suitable for publication in PLOS ONE. Congratulations! Your manuscript is now with our production department. 

Kind regards, 

on behalf of

Dr. Heidar Sharafi 

Academic Editor

PLOS ONE